# LEARNING TO CONTROL SELF-ASSEMBLING MORPHOLOGIES:
# A STUDY OF GENERALIZATION VIA MODULARITY

## ABSTRACT

Much of contemporary sensorimotor learning assumes that one is already given a complex agent (e.g., a robotic arm) and the goal is to learn to control it. In contrast, this paper investigates a modular co-evolution strategy: a collection of primitive agents learns to self-assemble into increasingly complex collectives in order to solve control tasks. Each primitive agent consists of a limb and a neural controller. Limbs may choose to link up to form collectives, with linking being treated as a dynamic action. When two limbs link, a joint is added between them, actuated by the 'parent' limb's controller. This forms a new 'single' agent, which may further link with other agents. In this way, complex morphologies can emerge, controlled by a policy whose architecture is in explicit correspondence with the morphology. In experiments, we demonstrate that agents with these *modular* and *dynamic* topologies generalize better to test-time environments compared to static and monolithic baselines. Project videos are available at https://doubleblindICLR19.github.io/self-assembly/.

## 1 INTRODUCTION

Only a tiny fraction of the Earth's biomass is composed of higher-level organisms capable of complex sensorimotor actions of the kind popular in contemporary robotics research (navigation, pick and place, etc). A large portion is primitive single-celled organisms, such as bacteria (Bar-On et al., 2018). Possibly the single most pivotal event in the history of evolution was the point when single-celled organisms switched from always competing with each other for resources to sometimes cooperating, first by forming colonies, and later by merging into multicellular organisms (Alberts et al., 1994). These modular self-assemblies were successful because they combined the high adaptability of single-celled organisms while making it possible for vastly more complex behaviours to emerge. Like many researchers before us (Murata & Kurokawa, 2007; Sims, 1994; Tu & Terzopoulos, 1994; Yim et al., 2000; 2007), we are inspired by the biology of multicellular evolution as a model for emergent complexity in artificial agents. Unlike most previous work however, we are primarily focused on modularity as a way of improving generalization to novel environmental conditions.

In this paper, we present a study of modular self-assemblies of primitive agents — "limbs" which can link up to solve a shared task. The limbs have the option to bind together by adding a joint that connects their morphologies (Figure 1a), and when they do so, they pass messages and share rewards. Each limb comes with a simple neural net that controls the torque applied to its joints. Linking and unlinking is treated as a dynamic action, so that the limb assembly can change shape during a single episode of the simulation. This setup has previously been explored in robotics as "self-reconfiguring modular robots" (Stoy et al., 2010). However, unlike prior work on such robots, where the control policies are hand-defined, we show how to *learn* the policies and study the generalization properties that emerge.

To make this problem computationally tractable, we do not allow the limb assemblies to form cycles in morphology. Limbs pass messages to their neighbors in this graph in order to coordinate behavior. All limbs share a common policy function, parametrized by a neural network, which takes the messages from adjacent limbs as input and outputs a torque to rotate the limb in addition to the linking/un-linking action. We call the aggregate neural network a Dynamic Graph Network (DGN)

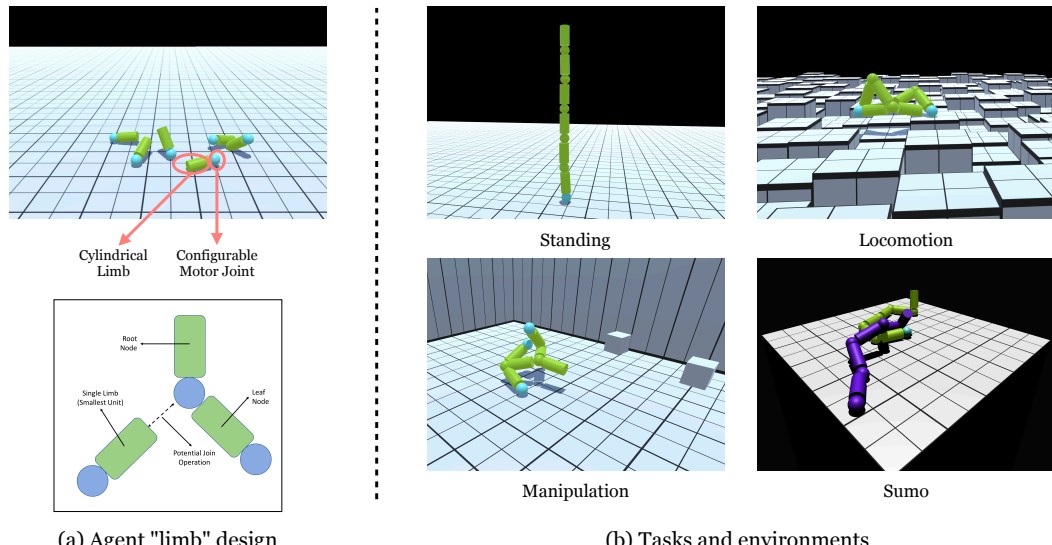

(a) Agent "limb" design  (b) Tasks and environments

Figure 1: We study the modular co-evolution of control and morphology where a collection of primitive agents self-assemble to form complex collectives to perform given tasks. (a) Each primitive agent is a limb containing a cylindrical body and a configurable motor. These limbs can connect with each other using the attached motor as a joint. (b) We illustrate our dynamic agents in four environments / tasks: standing up, locomotion, manipulation (pushing), and sumo wrestling. See project videos at https://doubleblindICLR19.github.io/self-assembly/.

since it is a graph neural network (Scarselli et al., 2009) that can dynamically change topology as a function of its own outputs.

We test our limb assemblies on four tasks: standing, locomotion, pushing and wrestling, shown in Figure 1b. We find that DGNs enable a single modular policy to control multiple possible morphologies, even those unseen during training. For example, a 6-limb policy, trained to build a 6-limb tower, can be applied at test time on 12 limbs, and results in a 12-limb tower. Not only are the policies robust to changes in number of limbs, they also generalize well to novel test-time environmental conditions, such as added wind, or new landscapes. These results together demonstrate that our modular and dynamic self-assembling agents have advantages toward generalization to new environments and tasks. Our main contributions are:

- Training primitive agents that self-assemble into complex morphologies to jointly solve control tasks.
- Formulating morphological search as a reinforcement learning problem, where linking and unlinking are treated as actions.
- Representing policy via a graph whose topology matches the agent's physical structure.
- Demonstrating that these self-assembling agents both train and generalize better than fixed-morphology baselines.

## 2  ENVIRONMENT AND AGENTS

Investigating the co-evolution of control (i.e., *software*) and morphology (i.e., *hardware*) is not supported within standard benchmark environments typically used for sensorimotor control, requiring us to create our own. We opted for a minimalist design for our agents, the environment, and the reward structure, which is crucial to ensuring that the emergence of limb assemblies with complex morphologies is not forced, but happens naturally.

**Environment Structure**  Our environment contains an arena where a collection of primitive agent limbs can self-assemble to perform control tasks. This arena is a ground surface equipped with

gravity and friction. The arena can be procedurally changed to generate a variety of novel terrains by changing the height of each tile on the ground (see Figure 1b). To evaluate the generalization properties of our agents, we generate a series of novels terrains. This include generating bumpy terrain by randomizing the height of nearby tiles, stairs terrain by incrementally increasing height of each row of tiles, hurdles terrain by changing height of each row of tiles, gaps terrain by removing alternate row of tiles, etc. Some variations also include putting the arena 'under water' which basically amounts to increased drag (i.e. buoyancy). We start our environment with a set of six primitive limb agents on the ground which can assemble to form collectives to perform complex tasks.

**Agent Structure**    All our primitive limb agents share the same simple structure: a cylindrical body with a configurable motor on one end. One end of the cylinder is free and the other end contains a configurable motor. The free-end of the limb can link up with the motor-end of the other limb, and then the motor acts as a joint between two limbs with three degrees of rotation. Hence, one can refer to the motor-end of the cylindrical limb as a *parent-end* and the free end as a *child-end*. Multiple limbs can attach their child-end to the parent-end of another limb, as shown in Figure 1(a), to allow for complex graph morphologies to emerge. The limb of the parent-end controls the torques of joint. The un-linking action can be easily implemented by detaching two limbs, but the linking action has to deal with the ambiguity of which limb to connect to (if at all). To resolve these modeling issues, we implement the linking action by attaching the closest limb within a small radius around the parent-node. If no other limb is present within the threshold range, the linking action has no effect.

The primitive limb agents are dropped in an environment to jointly solve a given control task. One key component of the self-assembling agent setup that makes it different from typical multi-agent scenarios (Wooldridge, 2009) is that if some agents assemble to form a collective, the resulting morphology becomes a new *single agent* and all limbs within the morphology maximize a joint reward function. The output action space of each primitive agent contains the continuous torque values that are to be applied to the motor connected to the agent, and are denoted by $\{\tau_\alpha, \tau_\beta, \tau_\gamma\}$ for three degrees of rotation. In addition to the torque controls, each limb can decide to attach another link at its parent-end, or decide to unlink its child-end if already connected to other limb. The linking and unlinking decisions are binary. This complementary role assignment of child and parent ends, i.e., parent can only link and child can only unlink, makes it possible to decentralize the control across limbs in a self-assembly.

In our self-assembling setup, each agent limb only has access to its local sensory information and does not know about other limbs. The sensory input of each agent includes its own dynamics, i.e., the location of the limb in 3-D euclidean coordinates, its velocity, angular rotation and angular velocity. Each end of the limb also has a trinary touch sensor to detect whether the end of the cylinder is touching 1) the floor, 2) another limb, or 3) nothing. Additionally, we also provide our limbs with a very simple point depth sensor that captures the surface height on a $9 \times 9$ grid around the projection of center of limb on the surface. One essential requirement to operationalize this setup is an efficient simulator to allow simultaneous simulation of several of these primitive limb agents. We implement our environments in the Unity ML (Juliani et al., 2018) framework, which is one of the dominant platforms for designing realistic games. For computational reasons, we do not allow the emergence of cycles in the self-assembling agents by not allowing the limbs to link up with already attached limbs within the same morphology. However, our setup is trivially extensible to general graphs.

## 3    LEARNING TO CONTROL SELF-ASSEMBLING MORPHOLOGIES

Consider a set of primitive limbs indexed by $i$ in $\{1, 2, \ldots, n\}$, which are dropped in the environment arena $\mathcal{E}$ to perform a given continuous control task. If needed, these limbs can assemble to form complex collectives in order to improve their performance on the task. The task is represented by a reward function $r_t$ and the goal of the limbs is to maximize the discounted sum of rewards over time $t$. If some limbs assemble to form a collective, the resulting morphology effectively becomes a single agent with a joint network to maximize the joint reward of the connected limbs. Further, the reward of an assembled morphology is a function of the whole morphology and not the individual agent limbs. For instance, in the task of learning to stand up, the reward is the height of the individual limbs if they are separate, but is the height of the whole morphology if those limbs have assembled into a collective. We now discuss our proposed formulation for learning to control these self-assembling agents.

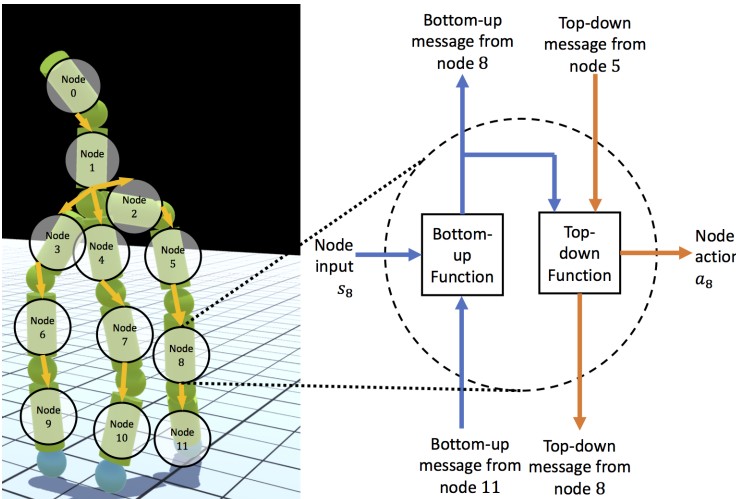

Figure 2: High-level visualization of our method. A set of primitive 'limbs' learn to self-assemble into morphologies where each limb is represented by a neural network linked via graph of physical edges. The inset on right shows the message-passing diagram for each node. Project videos at https://doubleblindICLR19.github.io/self-assembly/.

### 3.1 CO-EVOLUTION: LINKING/UNLINKING AS AN ACTION

To learn a modular controller policy that could generalize to novel setups, our agents must learn the controller jointly as the morphology evolves over time. The limbs should simultaneously decide which torques to apply to their respective motors, while taking into account the connected morphology. Our hypothesis is that if a controller policy could learn in a modular fashion over iterations of increasingly sophisticated morphologies (see Figure 3b), it could learn to be robust and generalizable to diverse situations. So, how can we optimize control and morphology under a common end-to-end framework?

We propose to treat the decision of linking and unlinking as additional actions of our primitive limb agents. The total action space $a_t$ at each iteration $t$ can be denoted as $\{\tau_\alpha, \tau_\beta, \tau_\gamma, \sigma_{link}, \sigma_{unlink}\}$ where $\tau_*$ denote the raw *continuous* torque values to be applied at the motor and $\sigma_*$ denote the *binary* actions whether to connect another limb at the parent-end or disconnect the child-end from the other already attached limb. This simple view of morphological evolution allows us to use ideas from learning-driven control, in particular, reinforcement learning (Sutton & Barto, 1998).

### 3.2 MODULARITY: SELF-ASSEMBLING AGENT AS A GRAPH OF LIMBS

Integration of control and morphology in a common framework is only the first step. The key question is how to model this controller policy such that it is modular and reuses information across generations of morphologies. Let $a_t^i$ be the action space and $s_t^i$ be the local sensory input-space of the agent $i$. One naive approach to maximizing the reward is to simply combine the states of the limbs into the input-space output all the actions jointly using a single network. Formally, the policy is simply $\vec{a}_t = [a_t^0, a_t^1 \dots a_t^n] = \Pi(s_t^0, s_t^0 \dots, s_t^n)$. This interprets the self-assemblies as a single monolithic agent, ignoring the graphical structure. This is the current approach to solve many control problems, e.g., Mujoco environments like humanoid (Brockman et al., 2016) where the policy $\Pi$ is trained to maximize the sum of discounted rewards using reinforcement learning.

In this work, we represent the policy of the agent via a graph neural network (Scarselli et al., 2009) in such a way that it explicitly corresponds to the morphology of the agent. Let's consider the collection of primitive agent limbs as graph $G$ where each node is denoted by to the primitive limb agent $i$. Two limbs being physically connected by a joint is analogous to having an edge in the graph. At a joint, the limb which connects itself via its parent-end acts as a parent-node in the corresponding edge, and the other limbs which connect to that joint via child-ends are child-nodes. The parent-node (i.e., the agent with the parent-end) controls the torque of the edge (i.e., the joint motor), as described in Section-2.

### 3.3 Dynamic Graph Networks (DGN)

Each primitive limb node $i$ has a policy controller of its own, which is represented by a neural network $\pi_\theta^i$ and receives a corresponding reward $r_t^i$ for each time step $t$. We represent the policy of the self-assembled agent by the aggregated neural network that is connected in the same graphical manner as the physical morphology. The edge connectivity of the graph is represented in the overall graph policy by passing messages that flow from each limb network to the other limbs physically connected to it via a joint. The parameters $\theta$ are shared across each primitive limb agent allowing the overall policy of the graph to be modular with respect to each node. However, recall that the agent morphologies are dynamic, i.e., the connectivity of the limbs changes based on policy outputs. This changes the edge connectivity of the corresponding graph network at every timestep, depending on the actions predicted by each limb controller network in the previous timestep. Hence, we call this aggregate neural net a *Dynamic Graph Network (DGN)* since it is a graph neural network that can dynamically change topology as a function of its own outputs in the previous iteration.

**DGN Optimization**  A typical rollout of our self-assembling agents during an episode of training contains a sequence of torques $\tau_t^i$ and the linking actions $\sigma_t^i$ for each limb at each timestep $t$. The policy parameter $\theta$ is optimized to jointly maximize the reward for each network limb:

$$\max_\theta \sum_{i=\{1,2...,n\}} \mathbb{E}_{\vec{a}^i \sim \pi_\theta^i}[\Sigma_t r_t^i] \tag{1}$$

We optimize this objective via reinforcement learning, in particular the policy gradient method PPO (Schulman et al., 2017).

**DGN Connectivity**  The topology is captured in the DGN by passing messages through the edges between individual network nodes. These messages allow each node to take into account its context relative to other nodes, and are supposed to convey information about the neighbouring policy network nodes in the graph. Since the parameters of these limb networks are shared across each node, these messages can be seen as context information that may inform the policy of its role in the corresponding connected component of graph. The aggregated flow through the whole graph can be encapsulated by passing these contextual messages in topological order (no cycles). One can either do a top-down pass, beginning from the root node (i.e., the node with no parents) to the leaf nodes, or do bottom-up pass, from leaves to root node. This idea is inspired from classical work on Bayesian graph networks where message passing is used for belief-propagation (Jordan, 2003). However, when the graph contains cycles, this idea can be easily extended by performing message-passing iteratively through the cycle until convergence, similar to loopy-belief-propagation in Bayesian graphs (Murphy et al., 1999). We now discuss these message-passing strategies:

*(a) Top-down message passing:* Instead of defining $\pi_\theta^i$ to be just as a function of state, $\pi_\theta^i : s_t^i \to a_t^i$, we pass each limb's policy network the information about its parent node as well. Formally, one can redefine $\pi_\theta^i$ as $\pi_\theta^i : [s_t^i, m_t^{p_i}] \to a_t^i$ where $p_i$ is the parent of node $i$. However, this also implies that each network node should pass context information as messages to its children networks for them to take it as input. So, we need to define $m_t^i$ which is the output of each node $i$, and which is passed as the input context message to all its children. We simply append this to the output of $\pi_\theta^i$. Thus, we finally define $\pi_\theta^i : [s_t^i, m_t^{p_i}] \to [a_t^i, m_t^i]$. If $i$ has no parents (i.e, root), a vector of zeros is passed in $m_t^{p_i}$. This is computed recursively until the messages reach the leaf nodes.

*(b) Bottom-up message passing:* In this strategy, messages are passed from leaf nodes to root, i.e., each agent gets information from its children, but not from its parent. Similar to top-down, we redefine $\pi_\theta^i$ as $\pi_\theta^i : [s_t^i, m_t^{C_i}] \to [a_t^i, m_t^i]$ where $m_t^i$ is the output message of policy that goes into the parent limb and $m_t^{C_i}$ is the aggregated input messages from all the children nodes, i.e, $m_t^{C_i} = \sum_{c \in C_i} m_t^c$. If $i$ has no children (i.e, root), a vector of zeros is passed in $m_t^{C_i}$. Messages are passed recursively until the root node.

*(c) Bottom-up then top-down message passing:* In this strategy, we pass messages both ways: bottom-up, then top-down. In the absence of cycles in graph, a one-way pass (either top-down or bottom-up) is sufficient to capture the aggregated information, similar to Bayesian trees (Jordan, 2003). Even though both-way message-passing is redundant, we still explore it as an alternative since it might help in learning when the agent grows too complex. This is implemented by dividing the policy into two

parts, each responsible for one direction of message passing, i.e., the parameters $\theta = [\theta_1, \theta_2]$. First the bottom-up message passing is formulated as $\pi_{\theta_1}^i : [s_t^i, m_t^{C_i}] \rightarrow m_t^i$ where the sensory input $s_t^i$ and input messages $m_t^{C_i}$ are used to generate outgoing messages to the parent node. In the top-down pass, messages from the parent are used, in addition with the agent's own message, to output its action: $\pi_{\theta_2}^i : [m_t^i, m_t^{p_i}] \rightarrow [a_t^i, \hat{m}_t^i]$ where $\hat{m}_t^i$ are the messages passed to the children nodes.

*(d) No message passing*: Note that for some environments or tasks, the context from the other nodes might not be a necessary requirement for effective control.In such scenarios, passing messages might creates an extra-overhead for training a DGN. Importantly, even with no messages being passed, the DGN framework still allows for coordination between limbs. This is because the control and morphology are still learned jointly in a modular mannner through the course of an episode i.e. the morphology and control in each timestep t depends explicitly on the physical morphology and the torques at previous timestep t 1. To implement the no message passing variant of DGN, we simply zero-out the messages $m_t^{p_i}, m_t^i$ at each timestep $t$. This is similar to a typical cooperative multi-agent setup (Wooldridge, 2009) where each limb makes its own decisions in response to the previous actions of the other agents. However, our setup differs in that our agents may physically join up, rather than just coordinate behavior.

## 4    IMPLEMENTATION DETAILS AND BASELINES

**Implementation Details:**    We use PPO (Schulman et al., 2017) as the underlying reinforcement learning method to optimize Equation 1. Limb policies are represented by fully-connected neural network and trained with a learning rate of $3e - 4$, discount factor of $0.995$ and entropy coefficient of $0.01$. Each episode is 5000 steps long at training and 1200 steps long at testing. Across all the tasks, the number of limbs at training is kept fixed to 6. Limbs start each episode disconnected and located just above the ground plane at random locations, as shown in Figure 3b. During generalization to novel scenarios, we experiment with changing the number of limbs to 12 or 3 to test the same policy without any further finetuning. All of our tasks require the agent to output continuous raw torque control values.

**Baselines**    We compare the role of the above four message passing strategies in DGN across a variety of tasks. Different strategies may work well in different scenarios. We further compare how well these dynamic morphologies perform in comparison to a learned monolithic policy for both dynamic and fixed morphologies. In particular, we compare to a (a) *Monolithic Policy, Dynamic Graph*: in this baseline, our agents are still dynamic and self-assemble to perform the task, however, their controller is represented by a single monolithic policy that takes as input the combined state of all agents and outputs actions for each of them. (b) *Monolithic Policy, Fixed Graph*: For each task, a hand-designed morphology is constructed from the limbs and trained using a single monolithic policy that takes as input the combined state of all agents and outputs the actions for all agents. The agents are not able to combine or separate This can be compared to a standard robotics setup in which a morphology is predefined and then a policy is learned to control it. Note that one cannot generalize *Monolithic Policy* baselines to scenarios where the number of limbs vary as it would change the action and state space of the policy.

For the *Fixed Graph* baseline, we chose the fixed morphology to be a straight line chain of 6-limbs (i.e., a linear morphology) in all the experiments including the task of standing up and locomotion. This linear-chain may be optimal for standing as tall as possible, but it is not necessarily optimal for learning to stand; the same would hold for locomotion. Further, note that, the best performing DGN variants also converges to linear-chain morphology (shown in Figure 3b and video results on the project website) to achieve the best reward in case of standing up task. Moreover, one can confirm that the locomotion task is also solvable with linear-morphology because one of the DGN ablation methods converged to a linear-morphology while doing well at locomotion (see video).

## 5    EXPERIMENTS: EMERGENT MORPHOLOGIES AND GENERALIZATION

We test the co-evolution of morphology and control across four tasks where self-assembling agents learn to: (a) stand up, (b) perform locomotion, (c) perform manipulation, and (d) fight in a sumo wrestling environment. There are two primary objectives of our investigation. The first is to determine

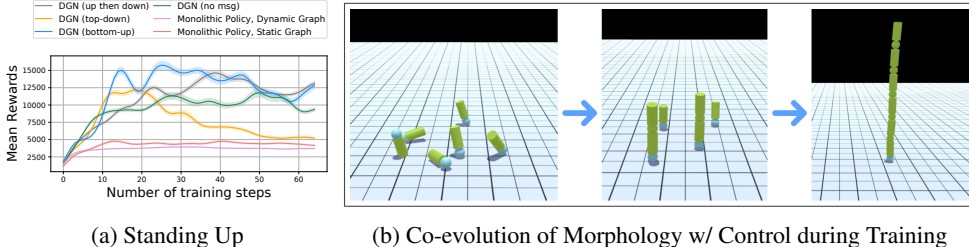

| (a) Standing Up | (b) Co-evolution of Morphology w/ Control during Training |

Figure 3: Training of self-assembling agents: (a) The training performance of different methods for joint training of control and morphology for the task of learning to stand up. The generalization performance of these policies across new scenarios is shown in Table 1. (b) The gradual co-evolution of controller as well as the morphology of self-assembling agents over the course of training.

| Environment | DGN | | | | Monolithic Policy | |
|---|---|---|---|---|---|---|
| | (up then down) | (top-down) | (bottom-up) | (no msgs) | (dynamic graph) | (fixed graph) |
| *Training Environment* | | | | | | |
| Standing Up | 15253 | 13486 | **17518** | 12470 | 4104 | 5351 |
| | | | | | | |
| *Zero-Shot Generalization* | | | | | | |
| More (2x) Limbs | 15006 (98%) | 14429 (107%) | **19796 (113%)** | 14084 (113%) | – | – |
| Fewer (.5x) Limbs | **11730 (77%)** | 9842 (73%) | 10839 (62%) | 9070 (73%) | – | – |
| Water + 2x Limbs | 16642 **(109%)** | 14192 (105%) | **16871** (96%) | 13360 (107%) | – | – |
| Winds | 14654 (96%) | 12116 (90%) | **16803** (96%) | 12560 **(101%)** | 3923 (96%) | 4531 (85%) |
| Strong Winds | 14727 (97%) | 13416 **(99%)** | **15853** (90%) | 12257 (98%) | 3937 (96%) | 4961 (93%) |

Table 1: Testing generalization for the *standing up* task. We show quantitative evaluation of the generalization ability of the learned policies. For each of the methods, we first pick the best performing model from the training run and then evaluate it on each of the novel scenarios without any further finetuning, i.e., in a zero-shot manner. We report first the score attained by the self-assembling agent and then report, in parenthesis, the percentage of training performance retained upon transfer. The higher the numbers, the better it is.

if such a modular co-evolution results in the emergence of complex self-assembling agents. The second is to evaluate if the emerged modular controller generalizes to novel scenarios.

## 5.1 TASK: STANDING UP

In this task, each agent's reward is proportional to the highest vertical point in its combined morphology, i.e., the limb assemblies should try to maximize their $Y$-axis height. Limbs have an incentive to self-assemble since the potential reward scales with the number of agents in the body, given that the agent can learn the controller for it. The learning process begins by six-limbs falling on the ground randomly, as shown in Figure 3b. In the beginning, each agent learns independently of others but these limbs learn to self-assemble to form a complex agent after training. Figure 3a compares different methods in terms of their performance on the task of standing as high as possible. We found that our DGN policy variants perform significantly better than the monolithic policies for the standing up task. In particular, the bottom-up and up-then-down message passing strategies attain the highest reward. To verify the implementation of our monolithic policy with fixed morphology, we show its ablation with varying number of limbs in Section A.1 in the supplementary.

However, the key question is whether the learned policy generalizes to novel scenarios. We investigate it by testing the learned policies without any further finetuning, i.e. zero-shot generalization, in novel scenarios: adding two times the number of limbs, reducing the number of limbs by half, increasing drag (i.e., 'under water') and number of limbs at the same time, and adding varying strength of random pushes-n-pulls (i.e., 'wind'). As the results in Table 1 show, DGN achieves similar performance as it did on the training environment, despite never having seen these scenarios before. Interestingly, the DGN variants seem to generalize better than the fixed-graph policies (last column). *Monolithic*

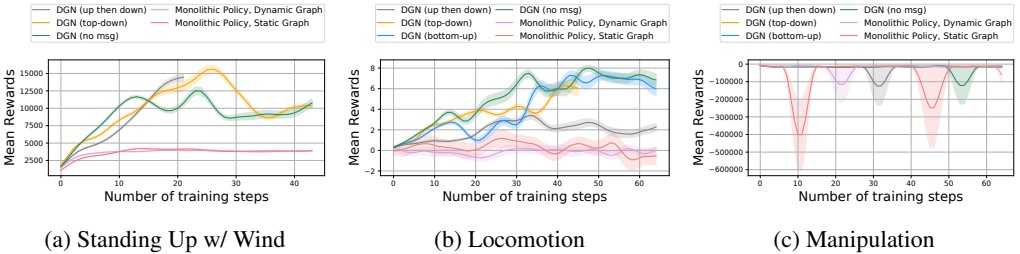

(a) Standing Up w/ Wind       (b) Locomotion       (c) Manipulation

Figure 4: Training self-assembling agents: We show the performance of different methods for joint training of control and morphology for three tasks: standing up in the presence of wind and random push-n-pulls (left), locomotion in bumpy terrain (center) and manipulation (pushing) of two objects (right). These policies generalize to novel scenarios as shown in respective tables.

| Environment | DGN | | | | Monolithic Policy | |
| --- | --- | --- | --- | --- | --- | --- |
| | (up then down) | (top-down) | (bottom-up) | (no msgs) | (dynamic graph) | (fixed graph) |
| *Training Environment* | | | | | | |
| Standing Up in Wind | 16339 | **18423** | – | 17237 | 4176 | 4500 |
| | | | | | | |
| *Zero-Shot Generalization* | | | | | | |
| (S)trong Winds | 15649 (96%) | **17384** (94%) | – | – | 4010 (96%) | 4507 **(100%)** |
| 2x Limbs + (S)Winds | **16250 (99%)** | 15351 (83%) | – | 15728 (91%) | – | – |
| Water + 2x(L) + (S)Winds | **17254 (106%)** | 17068 (93%) | – | 16592 (96%) | – | – |

Table 2: Testing generalization for the *standing up* task in the presence of random push-n-pulls (i.e. 'wind'). The best performing model from the training is evaluated on each of the novel scenarios without any further finetuning. The score attained by the self-assembling agent is reported first and then, in parenthesis, the percentage of training performance retained upon transfer. The bottom-up DGN failed due to some experimental error and will be reported in the final version of paper.

*policy* baselines cannot be generalized to more or fewer limbs due to the fixed action and state space. A better understanding of these results may be obtained by looking at the dynamically combining morphologies in the project video.

## 5.2   TASK: STANDING UP IN THE PRESENCE OF RANDOM PUSH-N-PULLS (WIND)

The task in this case is same as the previous one of learning to stand up. However, unlike in the previous subsection, here we also trained in the presence of random push-n-pulls (i.e., 'wind') with hope of making the learned morphologies even more robust. The training performance in Figure 4a show the superior performance of DGN with respect to the baselines. The generalization results, in Table 2, show that the DGN both-ways messaging passing variant is the most robust. This may be because in the presence of distractors, communication both ways can be helpful since a random force on a single limb affects all other attached limbs.

## 5.3   LOCOMOTION TASK

The reward function in this environment is defined as the distance covered by the agent along an axis, in particular, the limbs are rewarded is proportional to their velocity along the $X$-axis. The training environment is a bumpy terrain (shown in Figure 1(b)) and the training performance is shown in Figure 4b. Our DGN variants significantly outperform the monolithic baselines (see supplementary, Section A.1, for ablation). Interestingly, DGN variant with no message passing performs the best. Upon in-depth investigation, we found that it is possible to do well on this locomotion task with a large variety of morphologies, unlike the task of standing up where a tower is strongly preferable. Here, any morphology with sufficient height and forward velocity is able to make competitive progress in locomotion (see videos), and thus reducing message-passing to an unnecessary overhead. As discussed in Section 3.3, no message passing merely implies the absence of context to the limbs, but the DGN aggregated policy is still modular and jointly learned with the morphology over the episode.

| Environment | DGN | | | | Monolithic Policy | |
|---|---|---|---|---|---|---|
| | (up then down) | (top-down) | (bottom-up) | (no msgs) | (dynamic graph) | (fixed graph) |
| *Training Environment* | | | | | | |
| Locomotion | 3.91 | 6.87 | 8.71 | **9.0** | 0.96 | 2.96 |
| *Zero-Shot Generalization* | | | | | | |
| More (2x) Limbs | 4.01 **(103%)** | 4.29 (63%) | 5.47 (63%) | **9.19** (102%) | – | – |
| Fewer (.5x) Limbs | 3.52 (90%) | 4.49 (65%) | 6.64 (76%) | **8.2 (91%)** | – | – |
| Water + 2x Limbs | 2.64 (68%) | 3.54 (52%) | 6.57 (75%) | **7.2 (80%)** | – | – |
| Hurdles | 1.84 (47%) | 3.66 (53%) | **6.39 (73%)** | 5.56 (62%) | -0.77 (-79%) | -3.12 (-104%) |
| Gaps in Terrain | 1.84 (47%) | 2.8 (41%) | 3.25 (37%) | **4.17** (46%) | -0.32 (-33%) | 2.09 **(71%)** |
| Bi-modal Bumps | 2.97 (76%) | 4.55 (66%) | **6.62 (76%)** | 6.15 (68%) | -0.56 (-57%) | -0.44 (-14%) |
| Stairs | 1.0 (26%) | 4.25 (62%) | 6.6 (76%) | **8.59 (95%)** | -8.8 (-912%) | -3.65 (-122%) |
| Inside Valley | 4.37 **(112%)** | **6.55** (95%) | 5.29 (61%) | 6.21 (69%) | 0.47 (48%) | -1.35 (-45%) |

Table 3: Testing generalization for the *locomotion* task. The best performing model from the training is evaluated on each of the novel scenarios without any further finetuning. The score attained by the self-assembling agent is reported first and then, in parenthesis, the percentage of training performance retained upon transfer.

| Environment | DGN | | | | Monolithic Policy | |
|---|---|---|---|---|---|---|
| | (up then down) | (top-down) | (bottom-up) | (no msgs) | (dynamic graph) | (fixed graph) |
| *Training Environment* | | | | | | |
| Manipulation | -7985 | -7861 | -8482 | -9603 | -8773 | **-7725** |
| *Zero-Shot Generalization* | | | | | | |
| More (2x) Limbs | -14319 (-179%) | -14894 (-189%) | **-9969** (-118%) | -10879 **(-112%)** | – | – |
| Water + 2x Limbs | -10724 (-134%) | -13278 (-169%) | -12368 (-146%) | **-10362 (-108%)** | – | – |

Table 4: Testing generalization for the *manipulation* task. The score attained by the self-assembling agent is reported first and then, in parenthesis, the percentage of training performance retained.

We evaluate the learned policy without any further finetuning on several scenarios: more limbs, fewer limbs, more limbs under water, a terrain with hurdles of a certain height, a terrain with gaps between platforms, a bumpy terrain with a bi-modal distribution of bump heights, stairs, and an environment with a valley surrounded by walls on both sides. These environments are procedurally generated as discussed in Section 2. Across these novel environments, the modular policies learned by DGN tend to generalize better than the monolithic agent policies, as indicated in Table 3.

### 5.4    TASK: MANIPULATION OF TWO OBJECTS

The agents are dropped inside a room containing two objects and the goal is to decrease the distance between the objects, as shown in Figure 1(b). The reward for the agents is the negative distance between the objects, so as to encourage the behavior of pushing the blocks together. The training plots are shown in Figure 4c and the generalization results are shown in Table 4. This is a very hard task due to the sparse reward problem as agents only get reward if they move the block. Interestingly, the learned policies do not work well enough in this environment, and only learn to slightly move the blocks (see video). We believe this task requires more reward engineering than just the distance, and we will update the improved results in the final version.

### 5.5    TASK: SUMO WRESTLING BETWEEN TWO TEAMS

In this task, we divide the limbs into two teams of 6 limbs each and drop them into an arena to fight. Each team gets rewarded if any opponent limb falls out of the arena. The agents are trained via competitive self-play (Bansal et al., 2017; Tesauro, 1995). This is in contrast to the previous "single-team" tasks for self-assembling agents, i.e., standing, locomotion and manipulation. We present it as an additional result demonstrating the wider applicability of the method. However, it is non-trivial to measure the performance in self-play as the game is zero-sum, and rewards therefore do not increase over time. Instead, we refer the readers to the qualitative results in the video. The

policies learned by the self-assembling agents demonstrate some interesting behaviors, but there is a lot of room for improvement in future research. We will release these environments upon acceptance.

# 6 RELATED WORK

**Morphologenesis and self-reconfiguring modular robots** The idea of modular and self-assembling agents goes back at least to Von Neumman's *Theory of Self-Reproducing Automata* (Von Neumann et al., 1966). In robotics, such systems have been termed "self-reconfiguring modular robots" (Murata & Kurokawa, 2007; Stoy et al., 2010). There has been a lot of work in the modular robotics community in designing real hardware robotic modules that can be docked with each other to form complex robotic morphologies (Daudelin et al., 2018; Gilpin et al., 2008; Romanishin et al., 2013; Wright et al., 2007; Yim et al., 2000). Our main contribution is to approach this problem from a learning perspective, in particular deep RL, and study the resulting generalization properties.

A variety of alternative approaches have also been proposed to optimize agent morphologies, including genetic algorithms that search over a generative grammar (Sims, 1994), as well as directly optimizing over morphology parameters with RL (Schaff et al., 2018). One key difference between these approaches and our own is that we achieve morphogenesis via *dynamic actions* (linking), which agents take during their lifetimes, whereas the past approaches treat morphology as an optimization target to be updated between generations or episodes. Since the physical morphology also defines the connectivity of the policy net, our proposed algorithm can also be viewed as performing a kind of neural architecture search (Zoph & Le, 2016) in physical agents.

**Graph neural networks** Encoding graphical structures into neural networks has been used for a large number of applications, including quantum chemistry (Gilmer et al., 2017), semi-supervised classification (Kipf & Welling, 2016), and representation learning (Yang et al., 2018). The works most similar to ours involve learning control policies. For example, Nervenet (Wang et al., 2018) represents individual limbs and joints as nodes in a graph and demonstrates multi-limb generalization, just like our system does. However, the morphologies on which Nervenet operates are not learned jointly with the policy. hand-defined to be compositional in nature. Others (Battaglia et al., 2018; Huang et al., 2018) have shown that graph neural networks can also be applied to inference models as well as to planning. Many of these past works implement some variant of Graph Neural Networks (Scarselli et al., 2009) which operate on general graphs. Our method leverages the constraint that the morphologies can always be represented as a rooted tree in order to simplify the message passing.

# 7 DISCUSSION

Modeling intelligent agents as modular, self-assembling morphologies has long been a very appealing idea. The efforts to create practical systems to evolve artificial agents goes back at least two decades to the beautiful work of Karl Sims (Sims, 1994). In this paper, we are revisiting these ideas using the contemporary machinery of deep networks and reinforcement learning. Examining the problem in the context of machine learning, rather than optimization, we are particularly interested in modularity as a key to generalization, in terms of improving adaptability and robustness to novel environmental conditions. Poor generalization is the Achilles heel of modern robotics research, and the hope is that this could be a promising direction in addressing this key issue. We demonstrated a number of promising experimental results, suggesting that modularity does indeed improve generalization in simulated agents. While these are just the initial steps, we believe that the proposed research direction is promising and its exploration will be fruitful to the research community. To encourage follow-up work, we will release all code, models, and environments online once the paper is published.

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

# A SUPPLEMENTARY MATERIAL

## A.1 PERFORMANCE OF FIXED-GRAPH BASELINE VS. NUMBER OF LIMBS

To verify whether the training of *Monolithic Policy w/ Fixed Graph* is working, we ran it on standing up and locomotion tasks across varying number of limbs. We show in Figure 5 that the baseline performs well with less number of limbs which suggests that the reason for failure in 6-limbs case is indeed the morphology graph being fixed, and not the implementation of this baseline.

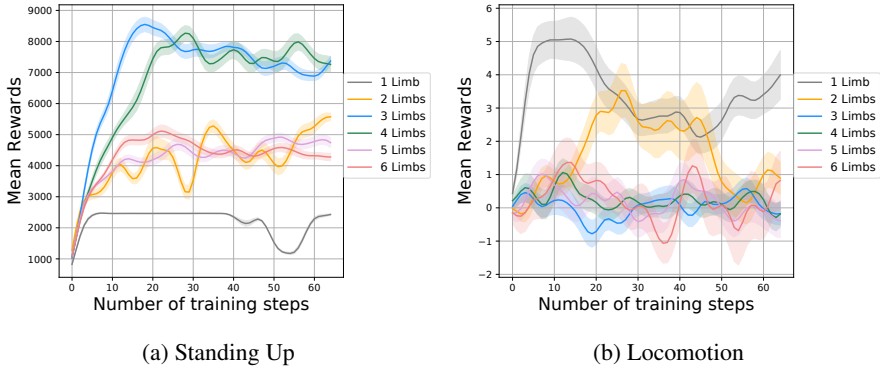

(a) Standing Up    (b) Locomotion

Figure 5: The performance of *Monolithic Policy w/ Fixed Graph* baseline as the number of limbs varies in the two tasks: standing up (left) and locomotion (right). This shows that the monolithic baseline works well with less (1-3 limbs), but fails with 6 limbs during training.

## A.2 GENERALIZATION OF LEARNED POLICIES AT DIFFERENT TRAINING INTERVALS

In this section, we show the generalization plots corresponding to the Tables 1, 2, 3, 4. To plot generalization, we pick the trained model from different training intervals and plot them across new environments without finetuning at all, in a zero-shot manner.

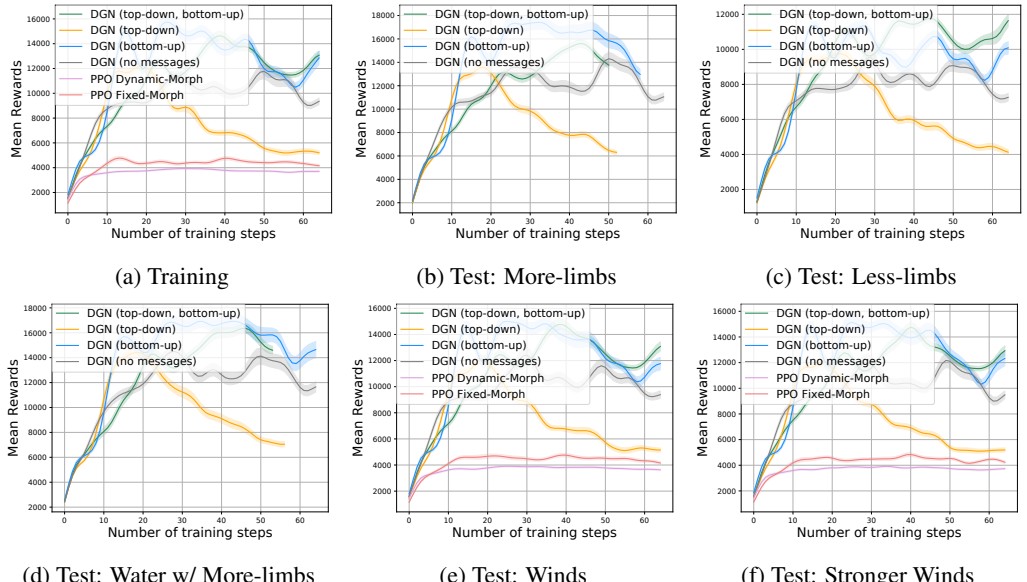

(a) Training    (b) Test: More-limbs    (c) Test: Less-limbs

(d) Test: Water w/ More-limbs    (e) Test: Winds    (f) Test: Stronger Winds

Figure 6: **Generalization for the task of Standing Up**: Performance of different methods across novel scenarios without any finetuning.

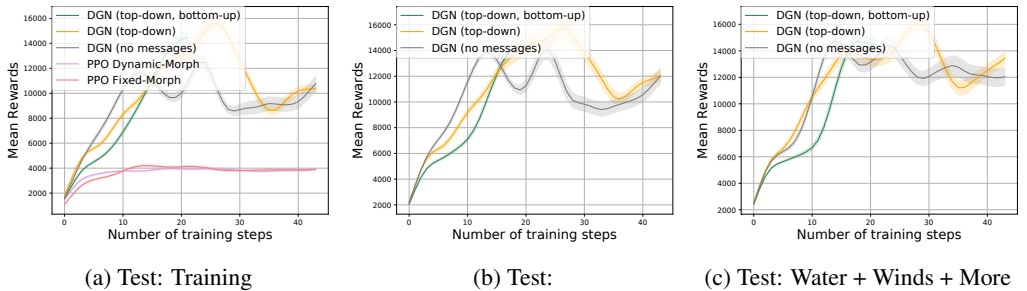

Figure 7: **Generalization for the task of Standing Up w/ Wind**: Performance of different methods across novel scenarios without any finetuning.

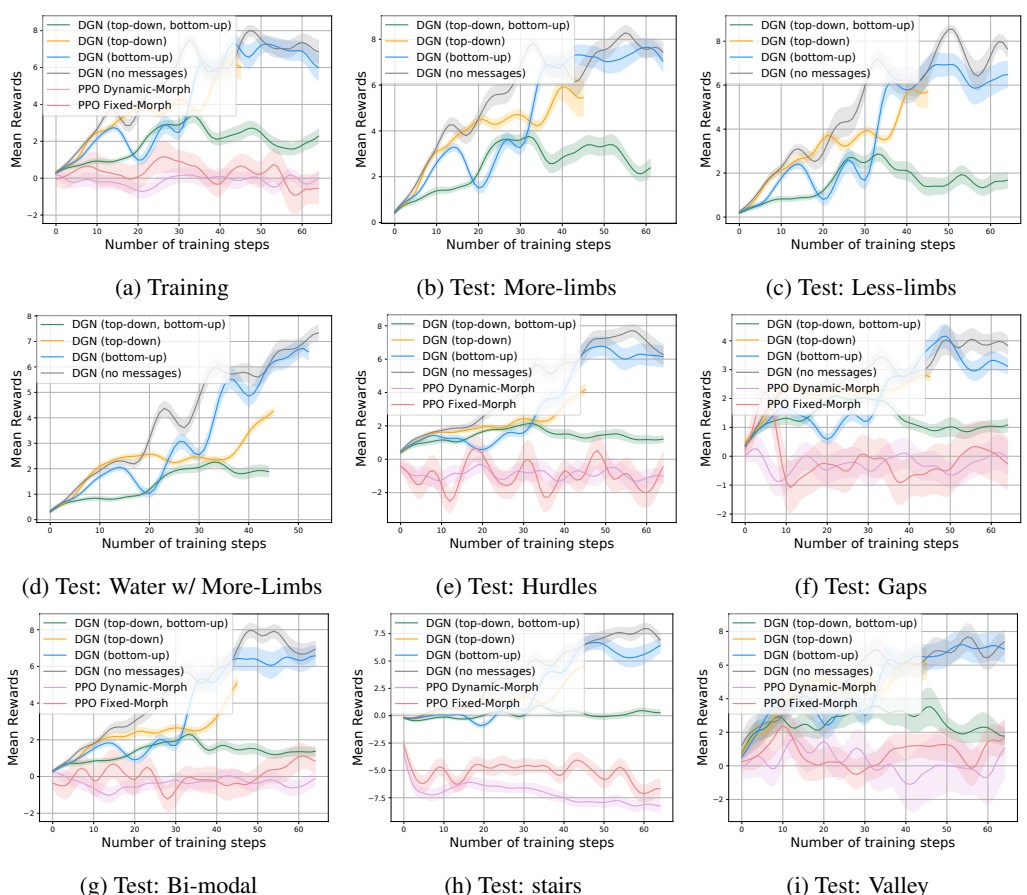

Figure 8: **Generalization for the task of Locomotion**: Performance of different methods across novel scenarios without any finetuning.

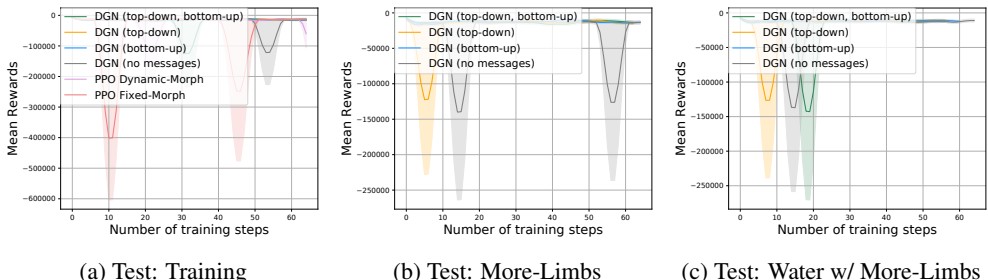

Figure 9: **Generalization for the task of Manipulation**: Performance of different methods across novel scenarios without any finetuning.

