# OpenReview forum: "Learning to control self-assembling morphologies: a study of generalization via modularity"
_ICLR.cc/2019/Conference_

### Official Review · AnonReviewer1 · 2018-10-30
**Interesting ideas and the setup, but virtually no details provided**

**Rating:** 4
**Confidence:** 4

**Review:**

Summary:
--------------
The paper considers the problem of constructing compositional robotic morphologies that can solve different continuous control tasks in a (multi-agent) reinforcement learning setting. The authors created an environment where the actor consists of a number of primitive components which interface with each other via "linking" and construct a morphology of a robot. To learn in such an environment, the authors proposed a graph neural network policy architecture and showed that it is better than the baselines on the proposed tasks.

I find the idea of learning in environments with modular morphologies as well as the proposed tasks interesting. However, the major drawback of the paper is the lack of any reasonable details on the methods and experiments. It's hard to comment on the novelty of the architecture or the soundness of the method when such details are simply unavailable.

More comments and questions are below. I would not recommend publishing the paper in the current form.


Comments:
----------------
- If I understand it correctly, each component ("limb") represents an agent. Can you define precisely (ie mathematically) what the observations and actions of each agent are?

- Page 4, paragraph 2: in the inline equation, you write that a sum over actions equals policy applied to a sum over states. What does it mean? My understanding of monolithic agents is that observations and actions must be stacked together. Otherwise, the information would be lost.

- Page 4, paragraphs 3-(end of section): if I understand it correctly, the proposed method looks similar to the problem of "learning to communicate" in a cooperative multi-agent setting. This raises the question, how exactly the proposed architecture is trained? Is it joint learning and joint execution (ie there's a shared policy network, observation and action spaces are shared, etc), or not? All the details on how to apply RL to the proposed setup are completely omitted.

- Is the topology of the sub-agents restricted to a tree? Why so? How is it selected (in cases when it is not hand-specified)?

- From the videos, it looks like certain behaviors are very unphysical or unrealistic (eg parts jumping around and linking to each other). I'm wondering which kind of simulator was used? How was linking defined (on the simulator level)? It would be nice if such environments with modular morphologies were built using the standard simulators, such as MuJoCo, Bullet, etc.


All in all, despite potentially interesting ideas and setup, the paper is sloppily written, has mistakes, and lacks crucial details.

---

> ### Author Response · Authors · 2018-11-21
> **[Authors' Response to R1] Updated paper with method, agent, environment details.**
>
> We thank you for the constructive feedback and are glad you found the modular morphologies and the proposed idea interesting.  Here we address your specific concerns. Please also see the "common response" posted separately.
>
> R1: "the major drawback of the paper is the lack of any reasonable details ..."
> =>  We apologize for the lack of detail. We have updated the paper with all the details about the method. The overall presentation of the paper has also been significantly improved.
>
> R1: " … define precisely (ie mathematically) what the observations and actions of each agent"
> => The output action space of each primitive agent contains the continuous torque values that are to be applied to the motor connected to the agent, and are denoted by $\{\tau_\alpha, \tau_\beta, \tau_\gamma\}$ for three degrees of rotation. In addition, the agent also outputs two binary actions $\{\sigma_{link}, \sigma_{unlink}\}$ which denote whether to connect another limb at the parent-end or disconnect the child-end from the other already attached limb.
>
> Each agent limb only has access to its local sensory information and does not know about other limbs. The sensory input of each agent includes its own dynamics, i.e., the location of the limb in 3-D euclidean coordinates, its velocity, angular rotation and angular velocity. Each end of the limb also has a trinary touch sensor to detect whether the end of the cylinder is touching 1) the floor, 2) another limb, or 3) nothing. Additionally, we also provide our limbs with a very simple point depth sensor that captures the surface height on a 9x9 grid around the projection of the center of limb on the surface.
>
> This text has been added to Section 2 of the updated draft.
>
>
> R1: "In the inline equation [Page 4, parag 2] … states, actions must be stacked together"
> => Yes, indeed it was a typo in the notation. Thank you. We have fixed it and explained it in Section 3.2 (first parag) of the updated draft.
>
> R1: "looks similar to the problem of "learning to communicate" in a cooperative multi-agent"
> => Our setup bears similarity to a multi-agent setup in the sense that each limb makes its own decisions in response to the previous actions of the other agents. However, our setup differs in that our agents may physically join up, rather than just coordinate behavior. One other key difference is that, when our agents assemble to form a collective, the resulting morphology becomes a new single agent and all limbs within the morphology maximize a joint reward function. We now mention this in Section 3.3 (last para) of updated draft.
>
> R1: ...how is it trained? joint? shared? how to apply RL?...
> => In our case, each limb has a learned controller of its own and the controller parameters are shared across limbs. The action space of each limb contains both torques as well as the commands to connect/disconnect.  This allows our self-assembling agent to jointly co-evolve their morphology along with the controller policy. DGN optimization and how it is performed using RL is described in the Section 3.3 (see Eq. 1) of the updated draft.
>
> R1: "Is the topology of the sub-agents restricted to a tree? Why so? How is it selected (in cases when it is not hand-specified)?"
> =>  We do not allow the emergence of cycles in agent topology for simple computational reasons: in case of cycles, one would have to perform message-passing iteratively through the cycle until convergence (similar to loopy-belief-propagation in Bayesian graphs). The topology is not hand-specified; we simply don't allow a limb to link up with already attached limbs within the same morphology. See Section 2 of the updated draft.
>
> R1: "It would be nice ... using the standard simulators, such as MuJoCo, Bullet, etc"
> => We implemented our environments in the standard Unity ML framework [Juliani et.al. 2018], which is one of the dominant platforms for designing realistic games and is efficient. We did try Mujoco briefly but found it hard to simulate lots of individual controllable limbs in parallel.
>
> R1: "How was linking defined (on the simulator level)?"
> R1: "certain behaviors are very unphysical or unrealistic eg parts jumping around and linking"
> => We implement linking action by attaching the closest limb within a small radius around the parent-node. If no other limb is present within the threshold range, the linking action has no effect. (see Section 2 of updated draft).  The linking mechanism is difficult to implement realistically in simulation and it makes things look somewhat unrealistic.
>
>
> We have also significantly improved the presentation quality of the overall paper, and would like to request the reviewer to take a second look at it. Thank you!

---

> > ### Comment · AnonReviewer1 · 2018-12-08
> > **Thanks for clarification.**
> >
> > I appreciate your effort to improve the paper.
> >
> > I still believe that a clear mathematical explanation is incomplete. How exactly is GCN integrated into PPO-based learning (end of page 5)? I'm not sure how exactly the messages are represented. A clear, step-by-step description of the algorithm (potentially with pseudocode) would help a lot.

---

> > > ### Author Response · Authors · 2018-12-08
> > > **Pseudo Code of the DGN Algorithm**
> > >
> > > Thank you for the suggestion of adding a pseudo-code of the overall algorithm. We provide the pseudo-code of our DGN algorithm below exactly as it is implemented in code line-by-line. Note that all these equations and parameters are already defined in Section 3.3. We will add this pseudo-code to the final version of the paper as open-review allows update. Hope this addresses your concern. We will also make our code publicly available.
> > >
> > > Looking forward to your reply!
> > >
> > > ----------------------------------------------------
> > > [Notation Summary as already defined in Section 3.3]
> > > ----------------------------------------------------
> > > For each node i:
> > >     First compute \pi_{\theta_1}^i (s_t^i, m_t^{C_i}) = m_t^i
> > >     Then compute \pi_{\theta_2}^i (m_t^i, m_t^{p_i}) = [a_t^i, \hat{m}_t^i]
> > > where:
> > >     s_t^i: observation state of agent limb i
> > >     a_t^i: action output of agent limb i: {3 torques, attach, detach}
> > >     m_t^{C_i}: aggregated message from children nodes input to agent i (bottom-up-1)
> > >     m_t^i: output message that agent i passes to its parent  (bottom-up-2)
> > >     m_t^{p_i}: message from parent node to agent i (top-down-1)
> > >     \hat{m}_t^i: message from agent i to its children
> > >     => \theta: {\theta_1, \theta_2}
> > >     => messages are 32 length floating point vectors.
> > >
> > > ----------------------------------------------------
> > > [Pseudo-code: Bottom-up, Top-down DGN]
> > > ----------------------------------------------------
> > > 1. Initialize parameters {\theta_1, \theta_2} randomly.
> > >    Initialize all message vectors {m_t^{C_i}, m_t^i, m_t^{p_i}, \hat{m}_t^i} to be zero.
> > >
> > > 2. Represent graph connectivity $G$ as [collection of nodes i, edges between nodes i]
> > >    Note: In the beginning, all edges are zeros, i.e., non-existent
> > >
> > > 3. Begin loop {for each time step t}
> > > 4.     Each limb agent i observes its own state vector s_t^i
> > > 5.     Begin loop {for each agent i}
> > > 6.         # Compute incoming child messages
> > >             m_t^{C_i} = 0
> > >             for each child node c of agent i in $G$:
> > >                 m_t^{C_i} += m_t^c
> > >
> > > 7.         # Compute message to parent p of agent i in $G$:
> > >             m_t^i := \pi_{\theta_1}^i (s_t^i, m_t^{C_i})
> > >
> > > 8.         # Compute action and messages to children of agent i:
> > >             a_t^i, \hat{m}_t^i := \pi_{\theta_2}^i (m_t^i, m_t^{p_i})
> > >
> > > 9.         # Execute morphology change as per a_t^i
> > >             if a_t^i[3]==attach:
> > >                 find closest agent j within distance d from agent i, otherwise j=NULL
> > >                 add edge (i,j) in $G$
> > >                 also make physical joint between (i,j)
> > >             if a_t^i[4]==detach:
> > >                 delete edge (i, parent of i) in $G$
> > >                 also delete physical joint between (i,j)
> > >
> > > 10.       # Execute torques from a_t^i
> > >              Apply torques a_t^i[0], a_t^i[1], a_t^i[2]
> > > 11.    End loop
> > >
> > > 12.    # Update message variables
> > >           Begin loop {for each agent i}
> > >               let p be parent of agent i in $G$
> > >               if p is NULL:
> > >                   set m_t^{p_i} to be zero
> > >               else:
> > >                   m_t^{p_i} = \hat{m}_t^i
> > >          End loop
> > >
> > > 13.    # Update graph and agent morphology
> > >          Find all connected components in $G$
> > >          Begin loop {for each connected component}
> > >               Begin loop {for each agent i in the connected component}
> > >                   reward r_t^i = reward of corresponding connected component (e.g. max height)
> > >               End loop
> > >          End loop
> > >
> > > 14. End loop
> > >
> > > 15. Update \theta = {\theta_1,\theta_2} to maximize joint discounted reward:
> > >         \max_{\theta} \mathbb{E} \sum_{agent i} [\sum_t r_t^i]
> > >     This is exactly same as an ordinary reinforcement learning objective.
> > >
> > >     => We optimize it by using off-the-shelf PPO to update \theta, i.e., {\theta_1,\theta_2} as follows:
> > >         let \vec{a}_t = [a_t^1, a_t^2.. a_t^n]
> > >             \vec{s}_t = [s_t^1, s_t^2.. s_t^n]
> > >             \hat{A}_t = advantage of discounted rewards, r_t = \sum_{agent i} r_t^i
> > >         PPO: \max_{\theta} \mathbb{E} [
> > >                             \hat{A}_t\frac{ \pi_\theta(\vec{a}_t|\vec{s}_t) }{ \pi_\thetaOld(\vec{a}_t|\vec{s}_t) }
> > >                           - \beta KL(\pi_\thetaOld(.|\vec{s}_t) || \pi_\theta(.|\vec{s}_t)) ]
> > >         Hyper-parameters: Section 4, Para 1
> > >
> > > 16. Repeat Step-3 to Step-15 until training converges
> > >
> > >
> > > ----------------------------------------------------
> > > [Pseudo-code: Bottom-up DGN]
> > > ----------------------------------------------------
> > > => Force set m_t^{p_i}=0 by removing Step-12.
> > >
> > >
> > > ----------------------------------------------------
> > > [Pseudo-code: Bottom-up DGN]
> > > ----------------------------------------------------
> > > => Force m_t^{C_i}=0 by removing Step-6.
> > >
> > >
> > > ----------------------------------------------------
> > > [Pseudo-code: No-message DGN]
> > > ----------------------------------------------------
> > > => Force m_t^{C_i}=0 by removing Step-6.
> > > => Force set m_t^{p_i}=0 by removing Step-12.

---

### Official Review · AnonReviewer2 · 2018-11-03
**An interesting idea of dynamical "self-assembly" but unclear implications of the proposed message passing**

**Rating:** 7
**Confidence:** 3

**Review:**

The paper describes training a collection of independent agents enabled with message passing to dynamically form tree-morphologies.  The results are interesting and as proof of concept this is quite an encouraging demonstration.

Main issue is the value of message passing
- Although the standing task does demonstrate that message passing may be of benefit. It is unclear in the other two tasks if it even makes a difference. Is grouping behavior typical in the locomotion task or it is an infrequent event?
  - Would it be correct to assume that even without message passing and given enough training time the "assemblies" will learn to perform as well as with message passing? The graphs in the standing task seem to indicate this. Would you be able to explain and perform experiments that prove or disprove that?
  - The videos demonstrate balancing in the standing task and it is unclear why the bottom-up and bidirectional messages perform equally well. I would disagree with your comment about lack of information for balancing in the top-down messages. The result is not intuitive.
  - Given the above, does message passing lead to a faster training?  Would you be able to add an experimental evidence of this statement?

---

> ### Author Response · Authors · 2018-11-21
> **[Authors' Response to R2] Discussing message-passing and it's value**
>
> We thank the reviewer for the constructive feedback and are glad that the reviewer found the results "interesting" and "quite an encouraging" demonstration of the proof of concept. Here we address your specific concerns. Please also see the "common response" posted separately.
>
> R2: Why does message passing help the standing task and not locomotion?
> R2: "Is grouping behavior typical in the locomotion task or it is an infrequent event?"
> => It is possible to do well on our locomotion task with a large variety of morphologies, unlike the task of standing up where a linear high tower is strongly preferrable. In locomotion, any morphology with sufficient height and forward velocity is able to make competitive progress (as also apparent in videos) hence the context provided by the messages is not as useful. So yes, linking-up is much less frequent in locomotion compared to standing up.
>
> However, no-message-passing merely implies the absence of context information provided to the limbs. The DGN aggregated policy is still modular and jointly learned with the morphology, two characteristics where we differ from more conventional agents, which usually have monolithic policies and fixed morphologies. We have clarified this in both Section 3.3 and Section 5.3.
>
> R2: "Would it be correct to assume that even without message passing and given enough training time the "assemblies" will learn to perform as well?"
> => Generally no. Each of our limb agent only receives its own local sensory information (clarified in Section 2 of updated draft) and does not have access to the global information about other limbs. Hence, in scenarios where the space of desired morphological structures is small, the role of message passing will be crucial.
>
> It might be possible that, in some scenarios, the self-assembling agents learn to "overfit" the training environment in a way that these contextual messages do not provide added benefit. But in such cases, the agent might have trouble generalizing to novel scenarios, such as, presence of random pushes-and-pulls (i.e., wind), adding more limbs etc.   One such example is shown in Table 2 where no-message-passing DGN works better than top-down DGN at training (row 1). However, it does not generalize as well and performs worse in novel environments with respect to the top-down DGN (row 3, 4).  A possible reason is that the presence of distractors (e.g. wind) may need the dynamic agent to change its morphology due to which the limb controllers can benefit from their context with respect to the remaining graph which is passed in messages.
>
>
> R2: "The videos demonstrate balancing in the standing task and it is unclear why the bottom-up and bidirectional messages perform equally well."
> => This is perhaps because we do not have any cycles in the morphology and hence one pass of message passing (either top down or bottom up) is sufficient to capture the information. This property is analogous to belief propagation in Bayesian Trees [Jordan et.al. 2003] where only directional pass is enough to obtain joint distribution; we draw this connection in Section 3.3 of the updated draft.
>
> R2: "I would disagree with your comment about lack of information for balancing in the top-down messages. The result is not intuitive."
> => We apologize the comment wasn't clear.  We think one of the reasons that top-down does not perform as well on the standing task could be because the controller policy for the limbs situated at the bottom of the tower has to be very precise to maintain the whole balance in comparison to the ones at the top of the tower. However, this is still a speculation and we would try to empirically investigate it in the final version of paper.
>
> R2: "does message passing lead to a faster training?  ... add an experimental evidence"
> => Our empirical observation suggests that the message passing is helpful in scenarios where the space of morphologies that perform well at a task is small. In such cases (e.g. standing), message passing indeed leads to faster training (as shown in Figures 3(a), 4(a) in the updated draft). However, message passing does not seem to have any effect on the training speed when many morphological structures can perform well at the same time (e.g., locomotion), as shown in Figure 3(b) of updated draft.
>
> Furthermore, we have significantly improved the presentation quality of the overall paper, and would like to request the reviewer to take a second look at it. Thank you!

---

### Official Review · AnonReviewer3 · 2018-11-04
**Collection of primitive agents is interesting, but**

**Rating:** 4
**Confidence:** 3

**Review:**

This paper investigates a collection of primitive agents that learns to self-assemble into complex collectives to solve control tasks.
The motivation of the paper is interesting. The project videos are attractive. However there are some issues:
1. The proposed model is specific to the "multi-limb" setting. I don't understand the applicability to other setting. How much generality does the method (or the experiment) have?

2. Comparison to other existing methods is not enough. There are many state-of-the-art RL algorithms, and there should be natural extension to this problem setting. I can not judge whether the proposed methods work better or not.

3. The algorithm is not described in detail. For example, detail of the sensor inputs, action spaces, and the whole algorithm including hyper-parameters are not explained well.

---

> ### Author Response · Authors · 2018-11-21
> **[Authors' Response to R3] Clarifying our contribution; Updated paper with details**
>
> We thank the reviewer for the constructive feedback and are glad that the reviewer found the general motivation "interesting" and the video results "attractive". Here we address your specific concerns. Please also see the "common response" posted separately.
>
> R3: The proposed model is specific to the "multi-limb" setting. I don't understand the applicability to other setting. How much generality does the method (or the experiment) have?
> => First, we would like to highlight that the multi-limb setting is quite broad: many real world robots are acyclic assemblies of rigid limbs.
> Second, we draw connections between the proposed approach and: a) multi-agent learning scenarios (Section 3.3), b) modular robotics (Section 6, para 1), c) automated robot design (Section 6, para 2), d) automated architecture search in neural networks (Section 6, para 2), and e) Bayesian Graphs (Section 3.3). We hope this work could serve as a stepping stone for future research on a range of agents (e.g., agents with soft bodies), or different materials for each limb, or more varied kinds of actuators.
>
> R3: "Comparison to other existing methods is not enough. There are many state-of-the-art RL algorithms, and there should be natural extension to this problem setting. I can not judge whether the proposed methods work better or not."
> => We apologise that it was not clear in our submitted draft, but our contribution is *not* about designing a new reinforcement learning algorithm (at least not in the sense of trying to create an optimizer that would be comparable to or compete with PPO, Q-learning, DDPG, etc.). Indeed, for all our experiments, we used a standard off-the-shelf RL method, PPO (Schulman et al.) as the underlying optimization tool. PPO was used to optimize our model and it was also used to optimize the baselines. We may just as well have used a different RL method, but choose to stick with PPO in the same way that papers on supervised learning may choose to use Adam as their optimizer without comparing to alternatives -- because it is standard and orthogonal to our contribution. The contribution of our approach is on the problem formulation and modeling side:
> (a) Formulating morphological search as a reinforcement learning problem, where linking and unlinking are treated as actions.
> (b) Representing policy via a graph whose topology matches the agent's physical structure.
>
> To further clarify, we have updated the legends of all the monolithic policy baselines in all the graphs, results and the tables to: (a) Monolithic Policy, Dynamic Graph and (b) Monolithic Policy, Fixed Graph. They are clarified in Sections 4, 5 in the updated draft of the paper. We have also added a concrete list of contributions to the end of Introduction (Section 1).
>
>
> R3: "detail of the sensor inputs, action spaces, and the whole algorithm … not explained well."
> =>   Thank you for valuable feedback. We have added full algorithm details in Section 3.3, and implementation details in Section 4 (first paragraph) in the updated draft of the paper.  The following sensor/action details have been added to Section 2 (last 2 paragraphs):
>
> Action Space: The output action space of each primitive agent contains the 3 continuous torque values (for 3 degrees of freedom) that are to be applied to the motor connected to the agent. In addition, the agent also outputs two binary actions which denote whether to connect or disconnect.
> Sensory Space: Each agent limb only has access to its local sensory information including: (a) own dynamics, i.e., the location of the limb in 3-D euclidean coordinates, its velocity, angular rotation and angular velocity; (b) a trinary touch sensor at each end to detect whether the end is touching the floor, another limb, or nothing; (c) a very simple point depth sensor that captures the surface height on a 9x9 grid around the limb.
>
>
> Furthermore, we have significantly improved the presentation quality of the overall paper, and would like to request the reviewer to take a second look at it. Thank you!

---

### Public Comment · (anonymous) · 2018-10-19
**Some deep issues with your results**

Dear authors,

Thanks for working on this problem which always takes us back to the classic Karl Sims results. But it seems like there are two very blatant issues in your experiments:

1.  There are no details on how you picked the fixed morphology for PPO. You have shown really bad training curves for the locomotion task, but it is quite well known now that any reasonable morphology can be trained to locomote when there are no bugs in the implementation of PPO. So, it seems like the fixed morphology was picked to make sure the baseline doesn't work.

2. It seems like in most of your experiments the message-passing doesn't matter at all, ie no-message passing baseline works pretty well. So, if all these individual limbs can just independently work to locomote efficiently, the need for the whole DGN architecture is quite questionable.

---

> ### Author Response · Authors · 2018-10-24
> **Authors' response to comments**
>
> We thank you for taking the time to read our draft. Our answers are as follows:
>
> 1. "Fixed-Morphology Baseline"
> => For the fixed-morphology baseline, we chose the morphology to be a straight line chain of 6-limbs (i.e., a linear morphology) in all the experiments including standing-up and locomotion. This linear-chain may be optimal for standing as tall as possible, but it is not necessarily optimal for learning to stand; same would hold for locomotion.
>
> => Note that DGN also converges to linear-chain morphology to achieve the best reward in case of standing-up task (e.g., see video results on the project website). Moreover, one can confirm that the locomotion task is also solvable with linear-morphology because one of the DGN ablation methods converged to a linear-morphology while doing well at locomotion.
>
> => The underlying PPO code is used off-the-shelf from a publicly available implementation (https://github.com/ikostrikov/pytorch-a2c-ppo-acktr) and is kept same across all methods in the graph without any change.
>
> => That being said, we were indeed surprised at baseline not performing too well and had been working on improving it. We recently found that it is hard to train fixed-morphology baseline for 6-limbs while it works well with 4-limbs. However, in either case, it does not seem to generalize as well. We will include these latest findings in an updated draft of the paper.
>
> 2. "Role of Message-passing in DGN"
> => We would like to emphasize that the message-passing DGN works significantly better than the non-message passing variant in the standing-up task. For instance, there is a significant gap between blue-curve (message passing) and gray-curve (non-message passing) in Figure-1.
>
> => For the locomotion task, in particular, the graphs do indicate that the message-passing does not improve the performance. We investigated this issue in depth and found out that it is possible to do well on the current bumpy-terrain-locomotion task without making any complicated morphology. For instance, any morphology with sufficient height and forward velocity can make comparable progress. We are running experiments by making the terrain even harder to verify indeed whether it is easiness of the task or the overhead of message-passing that makes non-message passing DGN work as well or better in this case.
>
> 3. Other Clarification:
> => Finally, we would like to clarify that the generalization curves denote the performance of different training checkpoints of a model across novel setups without any further fine-tuning on those setups (i.e., zero-shot). Hence, the checkpoint, which performs the best at training, is the one that matters the most in generalization plots instead of the whole x-axis. An alternate and cleaner way to present these generalization results would be to show scores in a table for the best training checkpoint.
>
> Hope this clarifies the raised questions. We would update the submitted version as the rebuttal period starts with these clarifications and new results.

---

### Author Response · Authors · 2018-11-21
**[Authors' Common Response] Major update to paper; improved presentation and details.**

We thank the reviewers for their insightful and helpful feedback. We are glad reviewers found the general motivation of proposed task "interesting" (R1, R2, R3), the video results "attractive" (R3) and "as a proof of concept... quite an encouraging demonstration" (R2).   However, all reviewers were concerned about missing details in the method and experiment sections. We apologize for this lack of clarity.

Motivated by the reviewers' comments, we have done a major update of the paper, clarifying the experiments and hopefully addressing all the reviewers' questions and concerns.  Here we summarize the key changes we have made:

1) Improved overall presentation: updated introduction, environment/agents details, method section and discussion of results.
2) Added a list of contributions to the end of the introduction
3) Replaced generalization graphs with tables: we realized that showing the generalization results as plots was unnecessarily confusing. These plots showed zero-shot generalization performance at each *iteration* of training. However, what is more common is to pick the single best policy from training (the one that achieves highest training reward), then test how well it generalizes to new scenarios. In our revision, we report these numbers in Table 1. (For completeness, we have moved the original plots to the supplementary material.)

We will answer individual questions that the reviewers raised in the respective replies, and look forward to their follow-up advice.

---

> ### Author Response · Authors · 2018-12-08
> **Added Pseudo-code of the algorithm in reply to R1**
>
> Dear Reviewers:
>
> R1 suggested us to provide a pseudo-code of the overall algorithm so as to improve the understanding of the algorithm. We have provided the pseudo-code of our DGN algorithm below in reply to R1. We will add this pseudo-code to the paper as soon as open-review permits update. We will also make our code publicly available.
>
> Looking forward to your comments!
>
>
> Psuedo-Code in reply to R1: https://openreview.net/forum?id=B1lxH20qtX&noteId=BklxiCctkN

---

### Comment · Area_Chair1 · 2018-12-08
**Final thoughts?  R3?**

We are coming to the end of the discussion phase.
Thank you for the discussion around R1 comments.
Hearing back from R3 would be very useful.
We do realize that everyone's time is limited.
-- area chair

---

### Meta-Review · Area_Chair1 · 2018-12-14
**many missing details; strange physics;**

**Confidence:** 5
**Recommendation:** Reject

**Metareview:**

Strengths: A co-evolution of body connectivity and its topology mimicing control policy is presented.

Weaknesses: Reviewers found the paper to be lacking in detail. The importance of message passing in achieving the given results is clear on one example but not some others. Some reviewers had questions regarding the baseline comparisons.
The authors provided lengthy details in responses on the discussion board, but reviewers likely had limited time to fully reread the many changes that were listed.
AC:  The physics in the motions shown in the video require signficant further explanation. It looks like the ball joints can directly attach themselves to the ground, and make that link stand up. Thus it seems that the robots are not underactuated and can effectively grab arbitrary points in the environment. Also it is strange to see the robot parts dynamically fly together as if attracted by a magnet.  The physics needs significant further explanation.

Points of Contention: The R2 review is positive on the paper (7), with a moderate confidence (3).
R1 contributed additional questions during the discussion, but R2 and R3 were silent.

The AC further examined  the submission (paper and video).
The reviewers and the AC are in consensus regarding
the many details that are behind the system that are still not understood.  The AC is also skeptical
of the non-physical nature of the motion, or the unspecified behavior of fully-actuated contacts
with the ground.